# Spirochetosis detection in colon histopathology images via fine-tuning and boosting techniques using foundation models

**Agata Polejowska**                                    AGATA.POLEJOWSKA@RADBOUDUMC.NL
**Fazael Ayatollahi**                                    FAZAEL.AYATOLLAHI@RADBOUDUMC.NL
**Ayse Selcen Oguz Erdogan**                    AYSESELCEN.OGUZERDOGAN@RADBOUDUMC.NL
**Francesco Ciompi**                                    FRANCESCO.CIOMPI@RADBOUDUMC.NL
**Annemarie Boleij**                                    ANNEMARIE.BOLEIJ@RADBOUDUMC.NL
*Radboud University Medical Center, Nijmegen, The Netherlands*

## Abstract

Spirochetes are bacteria that can be found on the boundaries of colon epithelial tissue, causing several diseases ranging from spirochetosis, inflammatory bowel disease, to cancer. Despite their relevance, spirochetes often remain undetected in histological analysis. We propose the first computational pathology approach to characterize spirochetes, leveraging prior spatial knowledge to detect spirochetes in whole-slide images of colon polyps and biopsies, and differentiate these bacteria as belonging to normal or abnormal tissue. We focus on transfer learning by fine-tuning state-of-the-art computational pathology foundation models and by training an additional XGBoost classifier on downstream tasks.

**Keywords:** Human intestinal spirochetosis, bacteria, histopathology, foundation models, transfer learning, feature extraction, XGBoost.

## 1. Introduction

In the colon, the interaction between its epithelial layers and microorganisms plays a crucial role in gastrointestinal health. Intestinal spirochetosis, recognizable by a distinctive fuzzy layer, called a false brush border, in the colon epithelium, is often associated with conditions like inflammatory bowel disease, colon polyps (Fan et al., 2022); (Ahmed et al., 2022); (Lemmens et al., 2019), gut-brain axis related diseases (Yuan et al., 2023), and also tumor (Rathje et al., 2020). Therefore spirochetosis detection in colon histopathology (biopsies and resections) is clinically relevant. Using computational image analysis techniques, our work aims to assist pathologists in identifying these bacteria more efficiently. To the best of our knowledge, this is the first work on spirochetosis detection with artificial intelligence in digital pathology. We leverage general-purpose foundation models in computational pathology such as UNI (Chen et al., 2024) and Phikon (Filiot et al., 2023), adapting them for spirochetosis classification. Adaptation strategies include transfer learning paradigms like fine-tuning the entire model or only the classification head while keeping the remaining parts of the transformer frozen. Foundation models can also serve as feature extractors for weakly supervised learning (Wölflein et al., 2024). The integration of state-of-the-art machine learning methods, like eXtreme Gradient Boosting (XGBoost) (Chen and Guestrin, 2016), operating in the extracted features space has not been yet explored in the proposed setting. The primary purpose of this study is to compare transfer learning and boosting approaches in terms of their effectiveness in spirochetes detection task formulated as a patch-based classification using prior knowledge that spirochetes reside on the luminal epithelium

and borders of the colon tissue. Additionally, we explore the characterization of spirochetes as belonging to normal or abnormal colon tissue boundary conditions.

## 2. Methods

**Dataset** A total of 78 whole-slide images of colon tissue were labeled at slide-level as either *normal* or *abnormal* (tubular adenoma, sessile serrated lesion, hyperplastic polyps) tissue. Furthermore, a pathologist manually annotated the epithelial layer regions containing spirochetes using coarse polygon annotations (see Figure 1). From the annotated regions at the tissue edge, non-overlapping patches of size 224x224 pixels were extracted, and used to build datasets to characterize spirochetes across varying health states and resolutions via patch classification: two sets of patches labeled with *presence* or *absence* of spirochetes, and two sets of spirochetes patches labeled as normal vs. abnormal, extracted at both 0.25 (10000 patches) and 0.5 (5000 patches) $\mu$m/px. The goal of using separate datasets with two different image resolutions is to investigate the context impact on the models' performance. We obtained four datasets in total. Each patch was sampled based on the whole-slide image and its corresponding mask using WholeSlideData (van Rijthoven, 2023). The dataset for classifying spirochetes presence consists of patches extracted from detected tissue contours including edges and holes (Lu et al., 2021). Each dataset was split allocating 70% for training, 15% for validation, and 15% for testing. The classes in datasets are balanced. Features from patches were extracted using unaltered models running in inference mode.

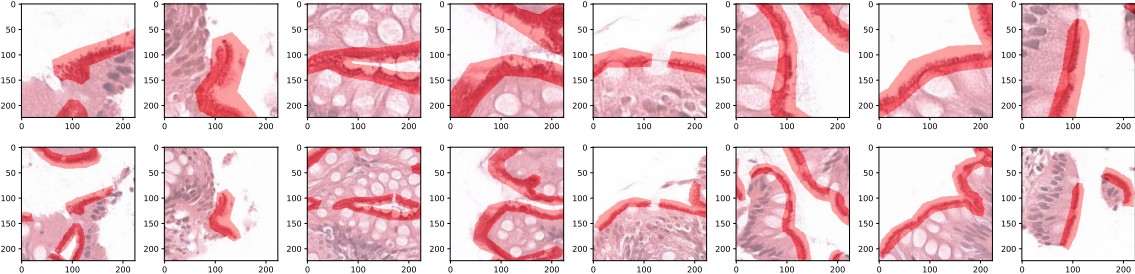

Figure 1: Example patches extracted with overlaid annotations of spirochetes. The rows correspond to spacings values 0.25 and 0.5 $\mu$m/px.

**Models and Evaluation** The Phikon and UNI foundation models were deployed as domain-specific transformers for fine-tuning and extracting features. For fine-tuning the entire model, we applied Low-Rank Adaptation (LoRA) (Mangrulkar et al., 2022). The frozen model had only the classification head fine-tuned using training data. Extracted features were used in the boosting algorithm. The XGBoost model operated in a feature space of dimensions dependent on the number of patches and the length of the created feature vector, 768 for Phikon or 1024 for UNI. The XGBoost was optimized by running hyperparameters tuning (Akiba et al., 2019) for 100 trials with F1 score maximization on the validation dataset as an objective. Patch classification was evaluated by comparing fine-tuned models versions against an XGBoost feature classifier. For fine-tuning, models were subjected to the same number of epochs, hyperparameters, data split and without additional

data augmentations to ensure a fair comparison. Models were benchmarked using a patch-level F1 score with weighted average in two binary patch classification tasks: detecting the presence of spirochetes and differentiating between normal and abnormal tissue containing spirochetes.

## 3. Results and Discussion

Table 1: F1 scores percentages calculated on test datasets across two spirochetes patch classification tasks on two spacing values, comparing Phikon and UNI models.

| Method | Spirochetes absence vs. presence | | | | Spirochetes normal vs. abnormal | | | |
|---|---|---|---|---|---|---|---|---|
| | F1 score (0.25) | | F1 score (0.5) | | F1 score (0.25) | | F1 score (0.5) | |
| | Phikon | UNI | Phikon | UNI | Phikon | UNI | Phikon | UNI |
| Fine-tuning entire model | 73.43 | 72.61 | 71.01 | 68.38 | 69.25 | 62.66 | 68.04 | 61.32 |
| Fine-tuning classification head | 80.09 | 74.36 | 84.26 | 73.96 | 74.37 | 65.42 | 80.99 | 64.90 |
| XGBoost optimized | 79.88 | **80.46** | 82.93 | **85.86** | 79.53 | **81.33** | 82.67 | **84.00** |

Based on Table 1, it can be observed that, in every case, fine-tuning the classification head or using XGBoost yields higher F1 scores than fine-tuning the entire model. This could be indicative of the need for more data or extended training periods for the entire model fine-tuning. Typically, XGBoost trained on UNI embeddings performs better than trained on Phikon embeddings. This might be attributed to the length of the feature vector as longer vector representations can encode additional and nonredundant signal. This can also explain that the difference between performance is larger for Phikon than for UNI for fine-tuning the classification head versus the entire model. For fine-tuning only the classification head as well as the entire model, Phikon delivers better performance across all experiments, this may mean that UNI needs more fitting time given the higher number of parameters in UNI than in Phikon. Whereas for classification effectiveness in the feature space, XGBoost applied to UNI embeddings delivers superior performance, especially with more context in a patch (0.5 $\mu$m/px spacing). Strong XGBoost performance observed for classifying UNI feature vectors can be due to its ability to efficiently leverage complex representations.

For *presence vs. absence*, performance consistency across spacings implies that resolution does not significantly impact the models' ability to detect spirochetes. This consistency is important in clinical settings, where variable resolutions are common. It suggests that presented models can be robust to varying magnifications in spirochetosis detection. Instead, for *normal vs. abnormal*, results suggest that a larger field of view and therefore more context may provide more information to discern abnormalities.

Overall, the conducted experiments show the feasibility of applying computational pathology foundation models through transfer learning and boosting techniques to the specified tasks. The proposed approach, using boosting on feature embeddings instead of vision transformer fine-tuning eliminates the necessity for GPU resources by using a CPU-based pipeline. Future work could encompass 1) the use of boosting to investigate feature importance for a patch selection scenario; 2) the combination of features extracted by different models, potentially gaining generalizability capabilities and thus improving performance in a given task; 3) research on the role of multiple resolutions on larger datasets.

## Acknowledgments

This work was supported by the HEREDITARY Project, as part of the European Union's Horizon Europe research and innovation programme under grant agreement No GA 101137074.

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
