# OpenReview forum: "Spirochetosis detection in colon histopathology images via fine-tuning and boosting techniques using foundation models"
_MIDL.io/2024/Short_Papers — MIDL 2024 Short Papers_

### Official Review · Reviewer_haPT · 2024-04-23

**Confidence:** 4
**Final Rating:** 3.5

**Review:**

The paper compares the classification performance of two foundation models that are finetuned to the task of spirochetosis detection in colon histopathology images. Three methodological approaches are compared for each model: fine-tuning the whole model, fine-tuning only the classification head, and fitting an XGBoost decision tree to the extracted features of the foundation model.

Strengths:
- The methodology is straight-forward and well explained.
- The intended usage of foundation models is verified. They can be used for small datasets with minimal fine-tuning effort.
- The classification performance of the XGBoost algorithm when trained on the extracted features is  very good.

Weaknesses:
- The paper confirms what was previously known: For small data quantities the usage of a traditional machine learning method such as XGBoost on the feature space of a foundation model yields very good performance (compare e.g. DINOV2 [1]).
- The authors do not discuss, why fine-tuning the classification head of the Phikon model leads better performance than for the UNI model. This needs to be improved.
- It is not clear, why the split between two different resolution levels is done.


[1] Oquab et al. (2023). DINOv2: Learning Robust Visual Features without Supervision. arXiv:2304.07193.

---

### Decision · Program_Chairs · 2024-04-26

Accept